

# Aerobic exercise improves verbal working memory sub-processes in adolescents: behavioral evidence from an N-back task

Yue Li[1,2], Fei Wang[1,2], Jingfan Li[1,2], Xing Huo[3] and Yin Zhang[1,2,4]

[1] Department of Psychology, Shaoxing University, Shaoxing, Zhejiang Province, China
[2] Center for Brain, Mind and Education, Shaoxing University, Shaoxing, Zhejiang Province, China
[3] Department of Physical Education, Shaoxing University, Shaoxing, Zhejiang Province, China
[4] Postdoctoral Research Station of Psychology, Henan University, Kaifeng, Henan Province, China

Corresponding author
Yin Zhang,
zhangyin0427@foxmail.com

## ABSTRACT

**Background:** Studies on the effects of aerobic exercise on working memory (WM) have mainly concentrated on the overall effects, yet there is little knowledge on how moderate intensity aerobic exercise impacts the sub-processes of verbal WM (VWM) in adolescents. To address this gap, two experiments were conducted to explore the influence of aerobic exercise on the maintenance and updating sub-processes of VWM.

**Methods:** In Experiment 1, a mixed experimental design of 2 (exercise habit: high *vs.* low) × 3 (memory load: 0-back *vs.* 1-back *vs.* 2-back) was used to compare VWM and its sub-processes in 40 adolescents. In Experiment 2, a 2 (group: intervention *vs.* control) × 3 (time point: pretest *vs.* 1st post-test *vs.* 18th post-test) × 3 (memory load: 0-back *vs.* 1-back *vs.* 2-back) mixed experimental design was used to investigate the acute and long-term effects of moderate intensity aerobic exercise on VWM and its sub-processes in 24 adolescents with low exercise habits.

**Results:** The results of Experiment 1 showed that VWM performance and its sub-processes in the high exercise habit group were better than those in the low exercise habit group. The results of Experiment 2 showed that the effects of the long-term exercise intervention were superior to those of the acute exercise intervention, and both were superior to the pretest. Meanwhile, it was found that aerobic exercise intervention had a greater effect size on the updating sub-process of VWM.

**Conclusion:** In conclusion, the results indicated that moderate intensity aerobic exercise could enhance the performance of VWM and its sub-processes in adolescents, and long-term intervention showed greater improvement effects compared to acute intervention, especially in the updating sub-process of VWM.

## INTRODUCTION

Working memory (WM) is a cognitive process that enables individuals to actively store and manipulate task-relevant information for a short period of time (*Baddeley, 2012*). It is an essential component of executive function, a higher-level cognitive ability involved in completing complex cognitive tasks (*Gross & Grossman, 2010*; *Diamond, 2013*).

*Baddeley (2000, 2012)* proposed a four-component model of WM that consists of a central executive system and two subsidiary slave systems: the phonological loop and the visuospatial sketchpad. Subsequent research has expanded this model to include long-term memory systems, which involve the semantic encoding of visual stimuli, contextual long-term memory, and language processing. The phonological loop is responsible for the storage and control of verbal information, such as the storage of voice and the articulatory control processing (*Camos & Barrouillet, 2014*). This loop holds verbal information for a short period of time, usually 1–2 s, and is represented by the speech structure (*Hitch, 2023*). Additionally, it allows for the conversion of written words into phonological codes to be stored in the phonological loop, such as phonological repetition and phonological conversion (*Baddeley & Hitch, 2019*). Verbal working memory (VWM) is a key factor in predicting reading comprehension (*Vernucci et al., 2021*).

The plasticity of WM suggests that it is subject to modification, especially in adolescents who demonstrate higher levels of malleability in their WM capacity (*Constantinidis & Klingberg, 2016*). Various methods, such as computerized training (*Diamond & Lee, 2011*; *Lin et al., 2018*) and video games (*Anderson-Hanley et al., 2014*), have been utilized to enhance WM in adolescents. However, these approaches often entail prolonged sedentary periods, which can be detrimental to their physical well-being (*Weiss et al., 2011*). As an alternative, aerobic exercise has been proposed as an enjoyable and sustainable means of improving WM in adolescents (*Chen et al., 2014*).

The brain plasticity hypothesis suggests that cognitive enhancements resulting from exercise stem from the brain's adaptability to movement-induced modifications (*Pedersen, 2019*; *Hill et al., 2023*). Research consistently shows that exercise can positively impact the hippocampal volume in adolescents, a critical region for learning and memory (*Curlik & Shors, 2013*; *Khan & Hillman, 2014*). Additionally, exercise is linked to elevated levels of brain-derived neurotrophic factor (BDNF) (*Ferris, Williams & Shen, 2007*; *Goekint et al., 2008*), a key regulator of aerobic exercise effects on hippocampal neuroplasticity and function (*Cotman, Berchtold & Christie, 2007*). Aerobic exercise is widely recognized as particularly influential in altering brain structure and function (*Stojiljković, Mitić & Sporiš, 2019*).

Aerobic exercise is widely recognized as an effective method for improving WM (*Affes et al., 2021*; *Wen, Yang & Wang, 2021*; *Zhu et al., 2021*). Moderate aerobic exercise intensity is generally considered safer and better tolerated (*Askim et al., 2014*). There is evidence that moderate intensity exercise increases BDNF (*Morais et al., 2018*), enhances cerebral blood flow to specific regions (*Robertson et al., 2015*), and improves behavioral performance (*Ploughman et al., 2008*). Studies have demonstrated that moderate intensity aerobic exercise can significantly improve WM (*Mcmorris & Hale, 2012*; *Ríos et al., 2016*), and this effect is even more pronounced when the exercise is done over a longer period of time (*Martins et al., 2013*; *Weng et al., 2015*; *Zhu et al., 2021*). For instance, *Martín-Martínez et al. (2015)* found that a long-term training program considerably increased the VWM of adolescents enrolled in a moderately intense aerobic exercise. In a study by *Zhu et al. (2021)*, deaf adolescents engaged in moderately intense aerobic exercise, such as treadmill running. The WM of deaf children increased dramatically after a long-term

moderate intensity training, according to results from a verbal n-back task. Moreover, neuroimaging studies demonstrated heightened neuronal connectivity and activation in the WM network, involving parietal, occipital, and hippocampal regions, in individuals after aerobic exercise. Therefore, it can be inferred that engaging in moderate intensity aerobic exercise for an extended duration significantly enhances WM.

*Wang et al. (2018)* highlighted the division of WM into two specific sub-processes: maintenance and updating. Maintenance involves the active storage and accessibility of information (*Cohen et al., 1997*), while updating is responsible for manipulating and adjusting representations, replacing outdated information with new data, and discarding irrelevant messages. Research has demonstrated that WM's two sub-processes, updating and maintenance, are largely distinct (*Trutti et al., 2021*). Individuals with cognitive impairment may have intact maintenance capacity, but deficits in the updating sub-process of executive control, and vice versa (*D'Esposito & Postle, 2000*). Additionally, WM updating training significantly improves language fluency in individuals with motor aphasia (*Rende, Ramsberger & Miyake, 2002*). Cognitive function enhancements have been observed in the elderly (*Li et al., 2008*), adults (*McNab et al., 2009*; *Jaeggi et al., 2008*; *Westerberg & Klingberg, 2007*), and children (*Klingberg, 2010*) post-aerobic exercise. Therefore, exploring the potential impact of aerobic exercise on VWM, particularly in the maintenance or updating sub-processes, is crucial.

This study aimed to investigate whether aerobic exercise enhances WM in the maintenance or updating sub-processes. The n-back paradigm was employed to assess VWM and its sub-processes in adolescents, with heart rate serving as an indicator of exercise intensity. Participants engaged in moderate-intensity exercise exclusively to ascertain the specific sub-processes of WM that benefit from aerobic exercise training.

# EXPERIMENT 1: COMPARISON OF VWM AND ITS SUB-PROCESSES IN ADOLESCENTS WITH HIGH AND LOW EXERCISE HABITS

## Method

### Participants

Using GPower 3.1, the expected power value for the repeated measures ANOVA in this study was 0.8, requiring a minimum sample size of 26 participants at a significance level of $\alpha = 0.01$, assuming a medium effect size ($f = 0.25$). A total of 40 participants (20 males and 20 females) between the ages of 12–14 from a middle school in Shaoxing City, China, were recruited. Inclusion criteria comprised right-handedness, normal or corrected-to-normal vision, non-color blindness, and a normal body mass index (BMI). None of the participants had prior experience with similar experiments. Participants received appropriate compensation upon completion of the study. Informed consent was obtained from both participants and their parents prior to the study, with approval granted by the middle school. The parents provided written consent, while the participants expressed their consent through verbal assent. The study was conducted in compliance with ethical

standards and was approved by the Ethical Committee of Shaoxing University (No. YXRQ-2022-002).

## Materials and design

In Experiment 1, a mixed design of 2 (exercise habits: high *vs.* low) × 3 (memory load: 0-back *vs.* 1-back *vs.* 2-back) was utilized. Exercise habits were the between-subjects variable, while memory load served as the within-subjects variable. Participants were categorized into high and low exercise-habit groups based on physical exercise scores obtained from the Chinese Center for Disease Control and Prevention (CDC) Health-Related/Risk Behavior Survey of Chinese Urban Adolescents (HRRBS).

The HRRBS was used to evaluate the exercise habits of Chinese adolescents (*Ji, 2007*; *Fu et al., 2021*). Participants were asked "On how many days did you partake in at least 60 min of exercise, such as walking, running, basketball, swimming, biking, or mopping the floor?" The answers were scored on an eight-point scale, with "0 to 7 days" scoring 0 to 7, respectively. Adhering to the Healthy China Action (2019–2030) standards, engaging in exercise for at least 30 min, three or more times a week, was considered regular participation. In this study, participants who scored 0 to 3 were placed in the low exercise habit group, while those with scores of 4 to 7 were categorized into the high exercise habit group.

## Procedure

The procedure for the verbal n-back task was based on a study by *Lytle, Hammer & Booth (2020)*. The experimental materials, consisting of nine capital English letters (A, B, C, E, H, K, N, P, and Z), were presented by E-prime 2.0 software and the three tasks were randomly performed for a duration of approximately 20 min. The participant's eyes were approximately 60 cm away from the screen. Participants were instructed to press the keys accurately and respond promptly.

In the modified n-back task, uppercase letters were utilized as the variable. Participants were tasked with different conditions: the 0-back required identifying if the current stimulus matched the first trial, the 1-back involved comparing the current stimulus to the previous trial, and the 2-back necessitated comparing the current stimulus to the one presented two trials earlier (Fig. 1).

## Data analysis

A two-factor repeated measures analysis of variance (ANOVA) was used to analyze the accuracy and response time (RT) respectively. The analysis factors included exercise habit (high *vs.* low) and memory load (0 *vs.* 1 *vs.* 2-back). All analyses were conducted using SPSS 26.0, with *P* values adjusted by the Greenhouse-Geisser method and partial eta-squared ($\eta_p^2$) calculated as the effect size. Data points beyond M ± 2.5 SD for the verbal 0 *vs.* 1 *vs.* 2-back task were excluded from the analysis. The updating sub-process in VWM was operationalized as the 1-back RT minus the 0-back RT, and the maintenance sub-process in VWM was defined as the 2-back RT minus the 1-back RT.

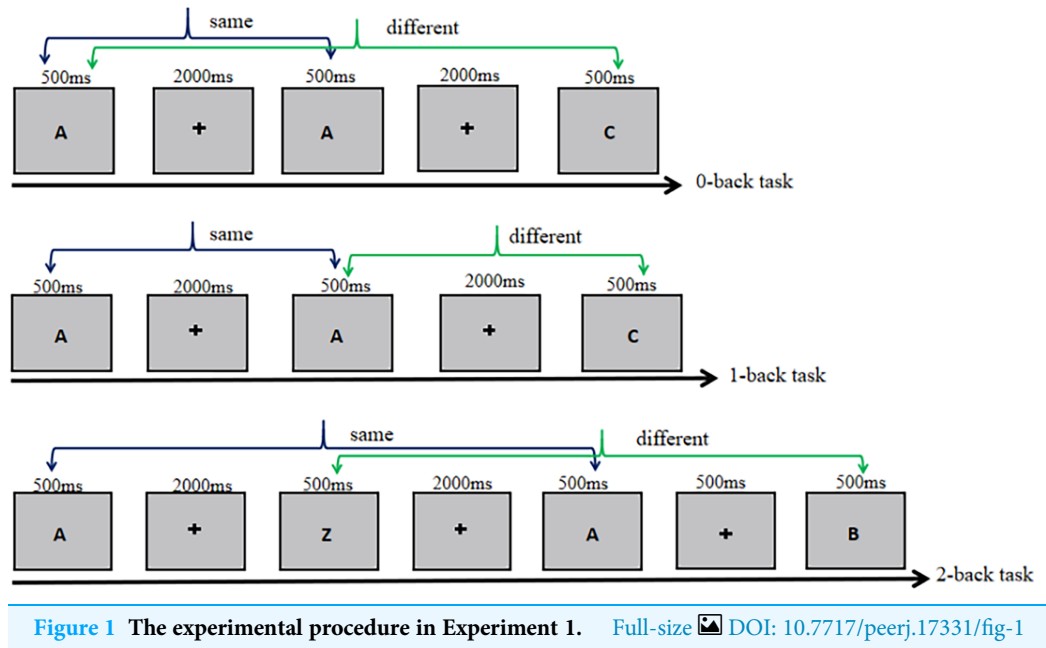

**Figure 1 The experimental procedure in Experiment 1.**

## Results

### The relationship between exercise habits and WM in adolescents

Table 1 shows that the high exercise habit group had superior accuracy and RT on the verbal memory load compared to the low exercise habit group. The results of a two-factor repeated measures ANOVA revealed that exercise habits had a significant effect on adolescents' memory load RT, with a main effect of $F (1,38) = 234.52$, $p < 0.001$, $\eta_p^2 = 0.86$. Results of the *post-hoc* analysis showed that the RT of the high exercise habit group was significantly lower than that of the low exercise habit group. The main effect of memory load level was significant ($F (2, 76) = 3,548.62$, $p < 0.001$, $\eta_p^2 = 0.99$). The interaction between exercise habit and memory load level was significant ($F (2, 76) = 230.28$, $p < 0.001$, $\eta_p^2 = 0.86$). Further simple effects revealed that in the verbal 0 *vs*. 1 *vs*. 2-back condition, the RT of the high exercise habit group was significantly lower than that of the low exercise habit group ($p < 0.001$) (Table 1).

### The influence of exercise habits on the sub-processes of VWM in adolescents

To determine the updating sub-process within VWM, the verbal 1-back RT was subtracted from the verbal 0-back RT, and for the VWM maintenance sub-process, the verbal 2-back RT was subtracted from the verbal 1-back RT. The results of a 2 × 2 repeated measures ANOVA revealed a significant main effect of exercise habits on the sub-processes of VWM in adolescents ($F (1, 38) = 316.94$, $p < 0.001$, $\eta_p^2 = 0.89$). *Post-hoc* analysis indicated that the RT for the VWM sub-process was significantly lower in the high exercise habit group *(M = 102.95 ± 2.78)* than in the low exercise habit group *(M = 172.94 ± 2.78)*. The interaction between exercise habit and sub-process of VWM was significant ($F (1, 38) = 7.89$, $p < 0.008$, $\eta_p^2 = 0.17$). Further simple effect analyzes revealed that the updating sub-process condition, the RT of the high exercise habit group *(M = 87.61 ± 5.20)*, was

**Table 1 The results of high/low exercise habit group on verbal 0/1/2-back task.**

| Task | | 0-back | Statistical test results | 1-back | Statistical test results | 2-back | Statistical test results |
|---|---|---|---|---|---|---|---|
| Reaction time (ms) | High exercise habit | 617.60 ± 14.10 | $t = 13.53$ $p < 0.001$ | 705.21 ± 53.66 | $t = 14.51$ $p < 0.001$ | 823.49 ± 53.18 | $t = 17.18$ $p < 0.001$ |
| | Low exercise habit | 887.30 ± 70.78 | | 1,056.84 ± 94.18 | | 1,233.18 ± 92.46 | |
| Accuracy rate (%) | High exercise habit | 0.98 ± 0.01 | $t = 5.43$ $p < 0.001$ | 0.97 ± 0.01 | $t = 6.79$ $p < 0.001$ | 0.79 ± 0.02 | $t = 13.53$ $p < 0.001$ |
| | Low exercise habit | 0.94 ± 0.01 | | 0.92 ± 0.01 | | 0.67 ± 0.03 | |

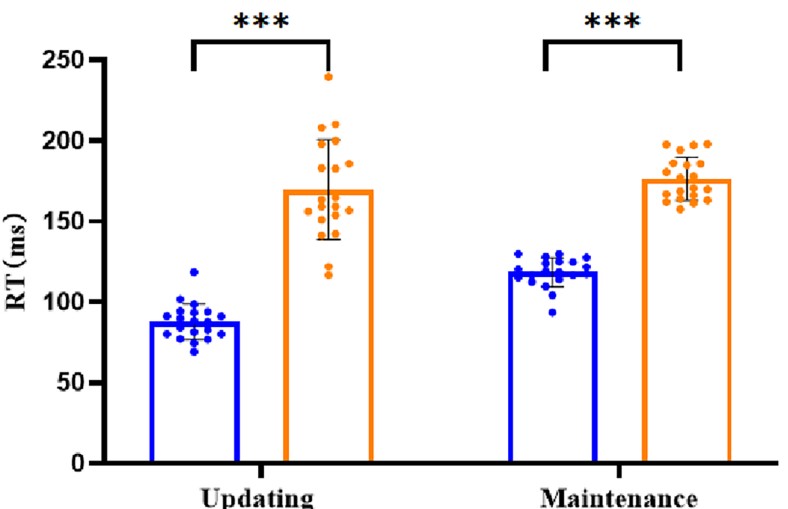

**Figure 2 The results of high and low exercise habit group on VWM sub-process (standard error).**

significantly lower than that of the low exercise habit group ($M = 169.55 ± 5.20$, $p < 0.001$). In the maintenance sub-process, the RT of the high exercise habit group ($M = 118.28 ± 2.55$) was significantly lower than that of the low exercise habit group ($M = 176.34 ± 2.55$, $p < 0.001$) (Fig. 2).

## Discussion

Experiment 1 was conducted to compare the VWM and its sub-processes between adolescents with high and low exercise habits. The results showed that the VWM of the high exercise habit group was superior to that of the low exercise habit group, and the maintenance and updating sub-process within VWM were also similar. This outcome was in line with our expectations.

Long-term exercise (high exercise habits) was found to have positive effects on VWM, which is in line with earlier research (*Van der Niet et al., 2016*). For instance, *Van der Niet et al. (2016)* utilized a digit span task to assess VWM in children and found that those who consistently engaged in physical exercise achieved higher scores. Adolescents with high exercise habits may engage in more verbal psycho-operational activities during their daily physical exercise routines than those with low exercise habits, as suggested by the Psychological Skills Hypothesis. This could lead to better reinforcement of verbal information, thus improving their capacity to store and update it.

To enhance the sub-processes within the VWM of adolescents and further promote their cognitive health development, it is necessary to explore ways to improve these sub-processes. Although studies have shown that moderate intensity aerobic exercise can improve VWM, it is still unknown which sub-processes of WM are affected. Experiment 2 utilized moderate intensity aerobic exercise as an exercise intervention for adolescents with low exercise habits in order to investigate the sub-processes in which it occurs. Our hypothesis was that a single exercise session (the first test) would have a positive effect on VWM and its sub-processes in adolescents. A long-term intervention (three times a week for 6 weeks) can promote VWM and its sub-processes in adolescents, especially the updating sub-process within WM.

## EXPERIMENT 2: EFFECT OF EXERCISE INTERVENTION ON VWM AND ITS SUB-PROCESSES IN ADOLESCENTS

### Method

#### Participants

Using Gpower_3.1 for the repeated-measures ANOVA applicable to this study, the expected power value was 0.8. The total sample size was at least 14 participants at a significance level of $\alpha = 0.01$ with a medium effect ($f = 0.25$). A total of 24 participants (12 males and 12 females) aged 13–15 from a middle school in Shaoxing City, China, participated in the study. Eligibility criteria included being right-handed, having normal or corrected-to-normal vision, not being color-blind, and having a normal body mass index (BMI). None of the participants had previously participated in similar experiments. Appropriate payment was given to the participants after the completion of the experiment. Prior to the experiment, informed consent was obtained from both the participants and their parents, and approval was granted by the middle school. The parents provided written consent, while the participants expressed their consent through verbal assent.

#### Materials and design

Experiment 2 employed a 2 (group: intervention *vs*. control) × 3 (time point: pretest *vs*. 1st post-test *vs*. 18th post-test) × 3 (memory load: 0-back *vs*. 1-back *vs*. 2-back) mixed experimental design. Participants were recruited and pre-tested one week before the intervention. The exercise intervention was conducted in the school playground, with participants engaging in a running queue. The intervention group began with 5 min of warm-up exercises, followed by 20 min of target heart rate zone exercise, and then a 5-min rest. The control group sat in two rows on the playground for 30 min in the same position

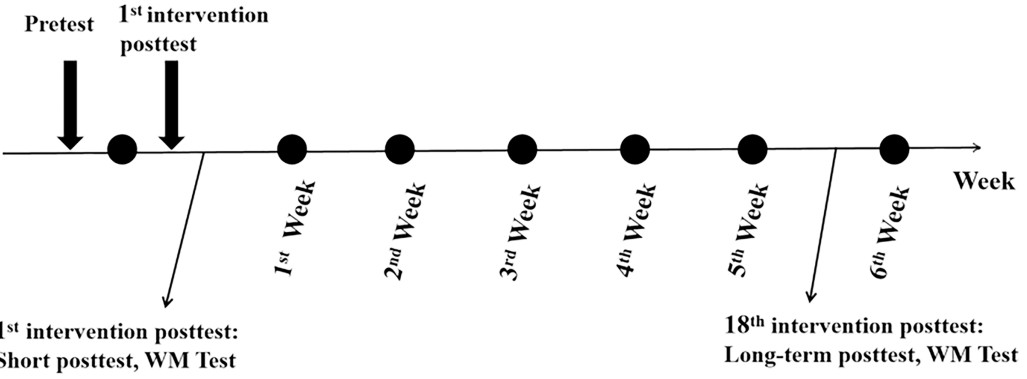

**Figure 3 The intervention procedure in Experiment 2.**

each time, and their heart rate was measured. All participants were required to maintain formation and rhythm during exercise to ensure that their heart rate was in the appropriate zone. After the intervention, all participants immediately engaged in computer operation in the computer room. To avoid practice and fatigue effects, the on-machine test was only performed after the first and eighteenth intervention sessions. The intervention exercise flowchart is shown in Fig. 3.

The exercise intervention program was based on the research from *Chen et al. (2016)* and *Schmidt et al. (2015)*. All participants in the exercise intervention group were offered an exercise program for 3 days per week for 6 weeks. The children's aerobic exercise load was of moderate intensity (maximum heart rate (MHR), 60–69%; MHR = 220–age), which has been shown to benefit cognition (*Zhu et al., 2021*). The participants' heart rate was monitored using a chest band heart rate exercise watch during the intervention, which included a sports watch and a chest band heart rate band. The measurement data were displayed in real time on the watch, which not only accurately measured the heart rate, but also monitored the heart rate zone in real time. If the participant's heart rate was not within the set target heart rate zone, a "beep" alarm would sound.

## Data analysis

The results were presented as mean ± standard deviation (M ± SD), and the statistical analysis was performed using SPSS 26.0. To investigate the impact of aerobic exercise intervention on VWM, a 2 (group: intervention/control) × 3 (time point: pretest/1st post-test/18th post-test) × 3 (memory load: 0-back/1-back/2-back) repeated measures ANOVA was conducted. *Post-hoc* comparisons were carried out in the event of a significant main effect, followed by analyses of significant interactions and subsequent simple effects. Separate repeated-measures ANOVAs were then used to assess the influence of aerobic exercise intervention on adolescents' VWM sub-processes, specifically maintenance and updating. *Post-hoc* analyses were performed through planned pairwise comparisons when both main and interaction effects were found to be significant. Corrections for *p* values were made using the Greenhouse-Geisser method, and the effect size was measured by calculating partial eta-squared ($\eta_p^2$).

**Table 2 Performance results for the time point segregated by group and verbal n-back task (mean ± standard deviation).**

| Group | Pre | | | Statistical test results | 1st | | | Statistical test results | 18th | | | Statistical test results |
|---|---|---|---|---|---|---|---|---|---|---|---|---|
| | 0-back | 1-back | 2-back | | 0-back | 1-back | 2-back | | 0-back | 1-back | 2-back | |
| Intervention | 703.35 ± 57.79 | 868.71 ± 39.47 | 1,109.41 ± 87.21 | $F(1,22) = 0.043$ $p = 0.837$ | 659.95 ± 36.84 | 781.11 ± 44.11 | 961.99 ± 42.43 | $F(1,22) = 32.71$ $p < 0.001$ | 565.02 ± 51.60 | 641.79 ± 56.53 | 755.67 ± 45.85 | $F(1,22) = 140.03$ $p < 0.001$ |
| Control | 708.77 ± 41.60 | 870.38 ± 32.71 | 1,114.16 ± 66.47 | $\eta_p^2 = 0.002$ | 701.27 ± 23.96 | 853.74 ± 36.93 | 1,083.40 ± 61.65 | $\eta_p^2 = 0.598$ | 670.83 ± 29.80 | 812.56 ± 33.83 | 1,036.19 ± 46.22 | $\eta_p^2 = 0.864$ |

## Result

### Effects of aerobic exercise intervention on the VWM memory load in adolescents

Table 2 showed that the intervention group had a better accuracy and RT on the time point than the control group. The findings of a $2 \times 3 \times 3$ repeated measures ANOVA indicated that group had a considerable influence on adolescents' RT in relation to the memory load, with a main effect of $F(1, 22) = 145.47$, $p < 0.001$, $\eta_p^2 = 0.87$. *Post-hoc* analysis revealed that the RT of the intervention group were significantly lower than the control group. The main effect of time point was significant ($F(2, 44) = 71.99$, $p < 0.001$, $\eta_p^2 = 0.77$). Results of the *post-hoc* analysis showed that the long-term exercise intervention post-test of RT was significantly lower than those of the acute exercise intervention post-test, and both were significantly lower than those of the pretest. The interaction between the group and time point was significant ($F(2, 44) = 26.58$, $p < 0.001$, $\eta_p^2 = 0.55$). Furthermore, simple effects analysis revealed that the pretest condition, the RT of the intervention and control groups, was not significant ($p = 0.837$). The first post-test (acute intervention) and the eighteenth post-test (long-term intervention) condition, the RT of the intervention groups, was significantly lower than that of the control groups ($p < 0.001$), and long-term intervention showed greater improvement compared to acute intervention. The summary statistics for the time point RT are shown in Table 2.

### Effects of intervention on the updating sub-processes in VWM

To investigate the effects of group (intervention, control) and time point (pretest, 1st intervention post-test, eighteenth intervention post-test) on the updating sub-process within VWM in adolescents, a $2 \times 3$ repeated measures ANOVA showed that the main effect of time point was significant ($F(2, 44) = 21.84$, $p < 0.001$, $\eta_p^2 = 0.50$). *Post-hoc* analysis indicated that the RT of the long-term exercise intervention ($M = 109.25 \pm 4.34$) was significantly lower than those of the acute exercise intervention ($M = 136.82 \pm 3.84$), and the RT was significantly lower than those of the pretest ($M = 163.49 \pm 7.28$). The main effect of group was significant ($F(1, 22) = 37.12$, $p < 0.001$, $\eta_p^2 = 0.63$). *Post-hoc* analysis indicated that the RT of the intervention group ($M = 121.10 \pm 3.58$) was significantly lower than those of the control group ($M = 151.94 \pm 3.58$). The interaction between the group and time point was significant ($F(2, 44) = 8.77$, $p < 0.001$, $\eta_p^2 = 0.29$). Furthermore, simple effects revealed that the intervention group condition, the RT of the updating sub-process

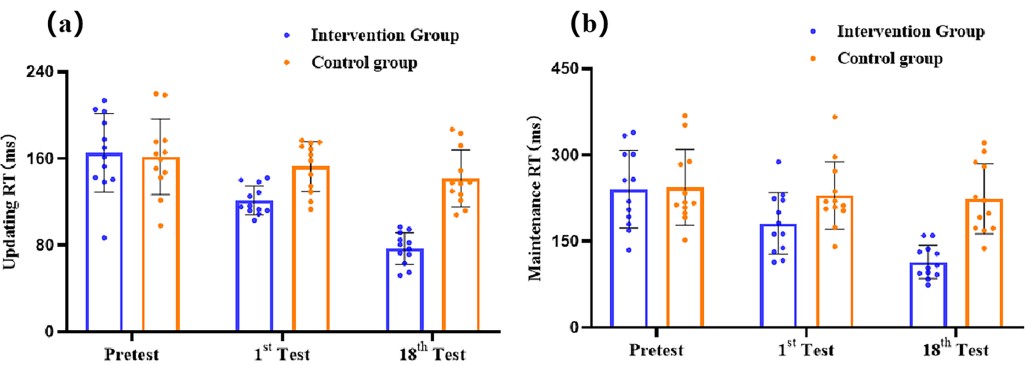

**Figure 4 The results of the intervention and control group updating (A) and maintenance (B) sub-processes within WM.**

of long-term exercise intervention post-test ($M$ = 76.76 ± 6.14), was significantly lower than those of the acute exercise intervention post-test ($M$ = 121.16 ± 5.43), and both were significantly lower than those of the pretest ($M$ = 165.36 ± 10.29), $p < 0.001$ (Fig. 4A).

### Effects of intervention on the maintenance sub-processes in verbal working

To investigate the effects of group (intervention, control) and time point (pretest, 1st post-test, 18th post-test) on the maintenance sub-process within VWM in adolescents, a 2 × 3 repeated measures ANOVA showed that the main effect of time point was significant ($F$ (2, 44) = 9.33, $p < 0.001$, $\eta_p^2 = 0.30$). *Post-hoc* analysis indicated that the RT of the long-term exercise intervention ($M$ = 168.76 ± 9.73) was significantly lower than those of the acute exercise intervention ($M$ = 205.27 ± 11.47), and the RT was significantly lower than those of the pretest ($M$ = 242.24 ± 13.62). The main effect of group was significant ($F$ (1, 22) = 17.79, $p < 0.001$, $\eta_p^2 = 0.45$). *Post-hoc* analysis indicated that the RT of the intervention group ($M$ = 178.49 ± 9.03) was significantly lower than those of the control group ($M$ = 232.36 ± 9.03). The interaction between group and time point was significant ($F$ (2, 44) = 4.95, $p = 0.012$, $\eta_p^2 = 0.18$). Further simple effects revealed that the intervention group condition, the RT of the maintenance sub-process within VWM long-term exercise intervention post-test ($M$ = 113.88 ± 13.75), was significantly lower than those of the acute exercise intervention post-test ($M$ = 180.89 ± 16.22), and the RT was significantly lower than those of the pretest ($M$ = 240.70 ± 19.26), $p < 0.001$ (Fig. 4B).

The results of Experiment 2 indicated that the VWM performance of the intervention group was better than that of the control group, with long-term exercise intervention being more effective than acute exercise intervention, both of which were superior to the pretest. The performance of the maintenance and updating sub-process of VWM is also the same. Comparing the effect size of moderate-intensity aerobic exercise intervention on the updating and maintenance sub-processes within VWM, it was found that the effect size of moderate intensity aerobic exercise intervention on the updating sub-process ($\eta_p^2$ = 0.50, $\eta_p^2 = 0.63$, $\eta_p^2 = 0.29$) was greater than that of the maintenance sub-process ($\eta_p^2 = 0.30$, $\eta_p^2 = 0.45$, $\eta_p^2 = 0.18$).

## Discussion

The results of Experiment 2 showed that moderate intensity aerobic exercise could enhance the performance of VWM and its sub-processes in adolescents. The effects of the long-term exercise intervention were superior to those of the acute exercise intervention, surpassing the pretest. Meanwhile, aerobic exercise intervention had a greater effect on the updating sub-process of VWM, which was in line with our expectations.

Research has demonstrated that moderate intensity aerobic exercise can improve VWM (*Wen, Yang & Wang, 2021*), and long-term moderate intensity exercise has a greater impact on VWM (*Martín-Martínez et al., 2015*; *Ríos et al., 2016*; *Zhu et al., 2021*). This experiment revealed that the VWM of adolescents improved as the number of exercise interventions increased, as evidenced by quicker RT. Moreover, the sub-processes' performances within WM also improved. We found that the improvement in VWM and its sub-processes was due to co-improvement. However, moderate intensity aerobic exercise had a more significant effect on the updating sub-process.

## GENERAL DISCUSSION

This research utilized the n-back paradigm to explore the influence of aerobic exercise on VWM sub-processes among adolescents. In Experiment 1, adolescents with high and low exercise habits were compared in terms of their WM performance, using three levels of memory load: 0, 1, and 2-back. Adolescents from the high exercise habit group had significantly better VWM performance than those from the low exercise habit group in situations of three memory load conditions, suggesting that long-term exercise can improve the updating and maintenance sub-process of WM. In Experiment 2, the performance of adolescents in the intervention and control group was assessed in the pretest, the first intervention post-test (acute intervention), and the eighteenth intervention post-test (long-term intervention) on three memory loads (0 *vs.* 1 *vs.* 2-back). The findings indicated that the long-term exercise intervention had a greater effect than the acute exercise intervention, and both had a greater effect than the pretest. It was discovered that exercise had a more advantageous effect on the updating component of VWM. This research concluded that aerobic exercise of moderate intensity can improve VWM and its components, regardless of the VWM load in adolescents. We found that long-term intervention had more of a beneficial effect than acute intervention, particularly in terms of the updating sub-process of VWM.

Previous research revealed that long-term exercise of moderate intensity significantly improves WM (*Affes et al., 2021*; *Wen, Yang & Wang, 2021*; *Zhu et al., 2021*), a finding that is supported by the results of the current investigation. Notably, however, some studies did not reproduce these findings. *Gothe et al. (2013)* conducted a study on female college students that included moderate-intensity aerobic exercise and yoga training. The findings of the study demonstrated that yoga training had a positive effect on participants' WM performance while acute moderate intensity aerobic exercise did not show any change. *Li et al. (2014)* reported that female college students did not experience any improvement in performance following acute moderate intensity aerobic exercise training, which contrasts with the findings of the present study. This discrepancy may be attributed to gender and

the duration of exercise training. Evidence suggests that gender may have an effect on participants' performance in WM tasks (*Blasiman & Was, 2018*; *Malagoli & Usai, 2018*). *Gothe et al. (2013)* and *Li et al. (2014)* both studied female participants, which could lead to biased results. Furthermore, their research showed that shorter exercise sessions had no effect on WM, and no significant changes were observed in participants' performance before and after exercise training. The cognitive stimulation hypothesis posits that if the duration of the intervention training is less than 20 min per session, it is unlikely that any positive psychological effects will manifest. Conversely, if the intervention lasts too long, it can lead to fatigue and injury. Thus, it can be concluded that a brief period of exercise may not have a substantial effect on an individual's WM performance.

There is still much debate about the impact of moderate-intensity aerobic exercise on WM capacity. This research indicated that moderate-intensity aerobic exercise was enhanced, regardless of the amount of memory required. However, different studies have produced divergent results, possibly due to the participants' initial levels. *Hsieh et al. (2017)* reported that moderate-intensity gymnastics training could enhance accuracy and reduce RT in a delay matching task, regardless of the length of the delay (3 s or 6 s). In addition, *Chen et al. (2016)* found that there was no improvement in the 0-back condition before and after aerobic exercise, but a significant difference was observed in the 2-back condition. *Yamazaki et al. (2018)* argued that the effect of aerobic exercise is contingent on the baseline performance and exercise intensity. Those with higher n-back scores showed less improvement as a result of exercise, whereas those with lower scores experienced more improvement. This study was conducted on individuals with low exercise habits, so it is likely that the enhancement of WM would have been greater if a different population was tested. To better understand the effects of aerobic exercise on WM, future studies should include different groups such as children, adults, seniors, and special populations.

Noting that the number of exercise interventions has risen, it is clear that adolescents' performance on the WM tasks (RT) improved in this study. Moreover, the performance of its sub-processes also grew. Long-term intervention was more advantageous than acute intervention, and both were superior to the pretest, suggesting that the improvement of WM (RT) by moderate intensity aerobic exercise may have stemmed from the improvement of its sub-processes. Moderate intensity aerobic exercise not only helps to retain information more effectively, but also enhances the speed of revising it. In accordance with cognitive control theory, cognitive-control can be split into two strategies: passive and active control (*Braver, 2012*). Passive control is a "late correction" approach, which obtains attention resources based on demand or the relevant plot without any intervention. Active control strategy is a type of "early selection" that consciously holds onto information pertinent to achieving a goal, thus ensuring the best cognitive performance.

WM is a fundamental element of cognitive control (*Baddeley, 1992*). To maximize the capacity to actively store data, sustaining an active control strategy can be beneficial for individuals' performance on WM tasks (*Speer, Jacoby & Braver, 2003*). Exercise is increasingly being acknowledged and implemented by researchers as an efficient way to strengthen active control strategies. *Ludyga et al. (2018)* conducted a longitudinal study

that revealed that adolescents who habitually take part in exercise (high exercise) had better WM maintenance (the capacity to store information actively) than those who did not exercise as often (low exercise). Studies by *Ludyga et al. (2020)* confirmed that children's WM can be improved by performing gymnastics activities for 8 weeks, as suggested by *Lin et al. (2021)*. They further contended that this phenomenon was due to the up-regulation of children's active control strategies.

Children and adolescents who do not frequently take part in physical activities (low activity) lack the ability to use active control strategies, which is likely due to a lack of attention resources and weaker memory capacity (*Braver, 2012*). Nevertheless, this can be improved with exercise intervention, leading to increased active control strategies to reduce the effect of distraction and increase the capacity to store information actively.

The Arousal Theory suggests that aerobic exercise can stimulate the autonomic nervous system, increase psychological arousal, and improve one's mental state. Evidence from *Chen et al. (2014)* suggests that moderate-intensity aerobic exercise can improve the updating sub-process of executive function in children's WM better than high-intensity exercise. Therefore, moderate aerobic exercise can help adolescents to improve their WM updating abilities.

## CONCLUSION

In conclusion, moderate-intensity aerobic exercise can improve the WM sub-processes in adolescents. Long-term intervention has more positive effects than a single session, particularly in terms of updating these sub-processes.

## ACKNOWLEDGEMENTS

The authors wish to thank the volunteers who participated in this study.

### Funding

This work was supported by the China Postdoctoral Science Foundation (2023M741006), the Zhejiang Postdoctoral Science Foundation (ZJ2023035), the Research Project of Zhejiang Province Association of Social Sciences (2024N067), and the Zhejiang Province Education Science Planning Program (2022SCG162). The funders had no role in study design, data collection and analysis, decision to publish, or preparation of the manuscript.

### Grant Disclosures

The following grant information was disclosed by the authors:
China Postdoctoral Science Foundation: 2023M741006.
Zhejiang Postdoctoral Science Foundation: ZJ2023035.
Research Project of Zhejiang Province Association of Social Sciences: 2024N067.
Zhejiang Province Education Science Planning Program: 2022SCG162.

### Competing Interests

The authors declare that they have no competing interests.

## Author Contributions

- Yue Li conceived and designed the experiments, performed the experiments, analyzed the data, prepared figures and/or tables, authored or reviewed drafts of the article, and approved the final draft.
- Fei Wang conceived and designed the experiments, performed the experiments, analyzed the data, authored or reviewed drafts of the article, and approved the final draft.
- Jingfan Li analyzed the data, prepared figures and/or tables, and approved the final draft.
- Xing Huo analyzed the data, prepared figures and/or tables, and approved the final draft.
- Yin Zhang conceived and designed the experiments, performed the experiments, analyzed the data, prepared figures and/or tables, authored or reviewed drafts of the article, and approved the final draft.

## Human Ethics

The following information was supplied relating to ethical approvals (*i.e.*, approving body and any reference numbers):

All procedures performed in studies involving human participants were in accordance with the ethical standards of the institutional and/or national research committee and with the 1964 Helsinki Declaration and its later amendments or comparable ethical standards. The study was approved by Human Research Institutional Review Board at Shaoxing University (the approval number: YXRQ-2022-002). Methods were carried out in accordance with relevant guidelines. Written informed consents were collected from participants.

## Data Availability

The raw measurements are available in the Supplemental File.

## Supplemental Information

Supplemental information for this article can be found online at http://dx.doi.org/10.7717/peerj.17331#supplemental-information.

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
