# Peer review of "Aerobic exercise improves verbal working memory sub-processes in adolescents: behavioral evidence from an N-back task"

_PeerJ, doi:10.7717/peerj.17331_

## Round 0.1 · original submission · Major Revisions

I have no further comments beyond what the reviewers have already mentioned.

**Language Note:** The review process has identified that the English language must be improved. PeerJ can provide language editing services - please contact us at [email protected] for pricing (be sure to provide your manuscript number and title). Alternatively, you should make your own arrangements to improve the language quality and provide details in your response letter. – PeerJ Staff

·

Basic reporting

The writing in English should be improved, mainly in the introduction.
Many sentences are seen without indicating the references in the introduction.
The figures and tables are adequate.

Experimental design

The research question is interesting, the need to conduct the research and address the knowledge gap is identified.
Follow international ethical standard.
The structure of the methods section could be presented as: design, sample size calculation, participants, outcomes, statistical analysis. In addition, all the data used to calculate the sample size should be integrated (e.g. statistical test and effect size used). This would facilitate reading.
Although they indicate that the exercise intervention is based on two studies, it would be appropriate to indicate the intervention in greater detail. They do not express whether it was aerobic exercise on a treadmill or on a stationary bike.

Validity of the findings

The findings are relevant since they carry out appropriate analyzes according to the subcategories of working memory, which provides deeper knowledge.
The conclusion is in line with the results.

Additional comments

In the title, it would be important to indicate the study design. It is the initial presentation to the reader.
Line 39: replace “better” with “greater”

In the introduction section I suggest indicating only information that is relevant to what has been developed in the research. Providing extra data that is not in accordance with what the researchers have done is not pertinent, therefore they could shorten the introduction so as not to divert the reader's attention to unnecessary data.

In the sentences that indicate “The upper layer” “the middle layer” should be changed by another concept since this could be interpreted as being limited to specific anatomical cortical brain structures instead of interpreting it as part of a working memory construct.

Line 68: you use physical activity and sport interchangeably. These are different concepts. Physical activity is any movement that involves energy expenditure and sports are activities that have rules.

Line 69-71: This idea could be developed more deeply to further support what they did.

Line 84-91: in this part there is data that is not necessary to mention. So much detail makes the reader get lost in the message you want to convey. I suggest eliminating the number of participants from these studies and their ages. It is understood that they give a framework of the participants, but what is relevant are the findings of the studies.

Line 128, 134, 238, 244: XX Middle School was indicated because the place is blinded? Or is it the real name of the place where it was applied?

Line 254-255: a “”beep” alarm will sound must be written in the past tense.

In tables 1 and 2 it would be a good option to indicate the statistically significant differences with symbols. It makes the table easier to read and makes it self-informative.

·

Basic reporting

no comment

Experimental design

A total of 40 participants were randomly selected from the students aged 12 to 14 at XX Middle School (20 males and 20 females) to participate in the study. The reason for selecting an equal number of males and females is to ensure the generalizability of the results and to better understand this age group of students.

Validity of the findings

.
.
The "Health-Related and Risk Behaviors Survey among Chinese Urban Adolescents" (HRRBAS) evaluates the exercise habits of Chinese adolescents. This evaluation is based on a wide range of research projects aimed at understanding the health-related and risk behaviors of Chinese adolescents.
.
.
The statistical analysis in sections 267-272 is too simple. Please provide a detailed description of the statistical methods used.
.

Additional comments

no comment

---

## Round 0.2 · accepted · Accept

The manuscript is ready for publication.

·

Basic reporting

There is a notable improvement in the manuscript.

Experimental design

The changes made allow us to better understand the research

Validity of the findings

The findings are clear

Additional comments

I have no additional comments. Good job and I appreciate that you integrated the suggestions. Without a doubt, it allows you to enhance your manuscript